# Variants Tagging *LGALS-3* Haplotype Block in Association with First Myocardial Infarction and Plasma Galectin-3 Six Months after the Acute Event

**DOI:** 10.3390/genes14010109

**Published:** 2022-12-29

**Authors:** Ana Djordjevic, Maja Zivkovic, Maja Boskovic, Milica Dekleva, Goran Stankovic, Aleksandra Stankovic, Tamara Djuric

**Affiliations:** 1Department of Radiobiology and Molecular Genetics, “Vinca” Institute of Nuclear Sciences—National Institute of the Republic of Serbia, University of Belgrade, 11001 Belgrade, Serbia; 2Department of Cardiology, University Clinical Centre “Zvezdara”, 11120 Belgrade, Serbia; 3Faculty of Medicine, University of Belgrade, 11000 Belgrade, Serbia; 4Department of Cardiology, Clinical Centre of Serbia, 11000 Belgrade, Serbia

**Keywords:** myocardial infarction, plasma galectin-3, *LGALS*-3 tag variants, haplotype, left ventricle

## Abstract

Galectin-3 is encoded by *LGALS-3*, located in a unique haplotype block in Caucasians. According to the Tagger server, rs4040064, rs11628437, and rs7159490 cover 82% (r^2^ > 0.8) of the genetic variance of this HapBlock. Our aims were to examine the association of their haplotypes with first myocardial infarction (MI), changes in left ventricular echocardiographic parameters over time, and impact on plasma galectin-3 and *LGALS-3* mRNA in peripheral blood mononuclear cells, both 6 months post-MI. The study group consisted of 546 MI patients and 323 controls. Gene expression was assessed in 92 patients and plasma galectin-3 in 189 patients. Rs4040064, rs11628437, rs7159490, and *LGALS-3* mRNA expression were detected using TaqMan^®^ technology. Plasma galectin-3 concentrations were determined by the ELISA method. We found that the TGC haplotype could have a protective effect against MI (adjusted OR 0.19 [0.05–0.72], *p* = 0.015) and that the GAC haplotype had significantly higher galectin-3 concentrations (48.3 [37.3–59.4] ng/mL vs. 18.9 [14.5–23.4] ng/mL, *p* < 0.0001), both in males and compared to the referent haplotype GGC. Higher plasma Gal-3 was also associated with higher NYHA class and systolic dysfunction. Our results suggest that variants tagging *LGALS-3* HapBlock could reflect plasma Gal-3 levels 6 months post-MI and may have a potential protective effect against MI in men. Further replication, validation, and functional studies are needed.

## 1. Introduction

Coronary artery disease (CAD), with atherosclerosis as the main underlying cause, is the number one killer worldwide, responsible for nearly 9 million deaths in 2019 [1]. Myocardial infarction (MI) is a prevalent manifestation of CAD with high morbidity and mortality rates. Major modifiable risk factors, as well as individual genetic heterogeneity, have been shown to increase MI susceptibility [2] and influence post-MI left ventricular (LV) remodeling [3,4].

In response to MI and consequential irreversible death of cardiomyocytes due to significant and sustained ischemia, inflammatory cells (predominantly neutrophils and macrophages) produce matrix metalloproteinases, growth factors, angiogenic factors, and pro-inflammatory factors, one of which is galectin-3 (Gal-3). Expressed Gal-3 is actively involved in the regulation of many aspects of inflammatory cell behavior. It has been confirmed that Gal-3 influences the formation, progression, and destabilization of atherosclerotic plaque by propagating vascular inflammation, binding lipopolysaccharides, and promoting the phenotypic modulation of vascular smooth muscle cells to a de-differentiated, synthetic state [5]. Accordingly, Gal-3 may be involved in atherothrombosis-induced CAD [6]. Sustained over-expression of Gal-3 is closely related to the induction of chronic and acute inflammatory conditions and inflammation leading to fibrosis of multiple organs, including myocardial fibrosis, as well as to the pathogenesis of cardiovascular (CV) remodeling [7,8]. Measurement of Gal-3 was recommended by the 2013 American College of Cardiology Foundation/American Heart Association (ACCF/AHA) clinical Guidelines as class IIb for management and risk prediction in heart failure (HF) [9].

Human Gal-3 is encoded by the *LGALS-3* gene, present on chr14q21-q22 in a unique, 300 kb long haplotype block in Caucasians. To date, few studies have analyzed the genetic background of *LGALS-3* mRNA and protein expression [10,11,12]. The two most investigated *LGALS-3* variants, non-synonymous rs4644 and rs4652, have previously been associated with circulating Gal-3 levels [10]. In the present study, we aimed to examine the genetic background of the aforementioned haplotype block using a bioinformatics tool, the Tagger server [13]. According to this method, we selected three tag variants, namely: rs4040064, rs11628437, and rs7159490, covering 82% (r^2^ > 0.8) of the genetic variance of this haplotype block. All three variants investigated affect *LGALS-3* mRNA expression, as presented by the GTEx project [14] and the FIVEx eQTL catalogue [15].

Previously, we found an association between two potentially functional variants, according to the RegulomeDB database [16], in the *LGALS-3* containing haplotype block, rs2274273 and rs17128183, with the risk for LV remodeling and with *LGALS-3* mRNA expression in peripheral blood mononuclear cells (PBMCs) from patients 6 months post-MI [17]. Recently, we examined haplotypes of the three tag variants mentioned above in association with carotid atherosclerosis, previous cerebrovascular insults, and *LGALS-3* mRNA in carotid plaque tissue [18].

In the present study we investigated: (1) whether tag variant (rs4040064, rs11628437, and rs7159490) haplotypes are associated with susceptibility to the first acute non-fatal MI and changes in echocardiographic parameters serving for assessment of LV function and structure within 6 months post-MI; (2) whether these haplotypes have an impact on plasma Gal-3 and relative *LGALS-3* mRNA expression in PBMCs, both in patients 6 months post-MI; (3) whether plasma Gal-3 correlates with age, body mass index (BMI), NYHA class of HF, and changes in echocardiographic parameters within 6 months post-MI in Serbian patients.

## 2. Materials and Methods

### 2.1. Study Population

The study consisted of 546 patients who survived their first acute MI and 323 healthy controls, all of whom were unrelated Caucasians of European descent from Serbia. The study enrolled 357 patients admitted to the Cardiology Clinic, Clinical Centre of Serbia, Belgrade, Serbia, from February to November 2013, and 189 patients admitted to the Coronary Care Unit in the Department of Cardiology, University Clinical Centre “Zvezdara”, Belgrade, Serbia, from December 2011 to September 2013, with symptoms of the first MI [19] as a consequence of CAD and referred for primary percutaneous coronary intervention (PCI). The exclusion criteria for all patients were: age over 70 years, history of previous MI or any other heart disease (valvular or congenital heart disease, cardiomyopathies), previous implantation of pacemaker or cardioverter-defibrillator, inability to undergo examination, or inadequate echocardiographic imaging. All the patients included in the study had a stenosis >70% in the infarct related coronary artery assessed by angiography, including single-vessel or multivessel CAD.

All biochemical analyses were performed at admission using standard laboratory procedures. Hypertension was defined as a systolic blood pressure ≥ 140 mmHg, a diastolic blood pressure ≥ 90 mmHg, or current treatment with antihypertensive drugs. Subjects with a fasting glucose level of ≥7.0 mmol/L, or who were taking insulin or oral hypoglycemic drugs, were characterized as having type 2 diabetes mellitus (T2DM).

A group of 189 patients recruited from the Coronary Care Unit in the Department of Cardiology, University Clinical Centre “Zvezdara”, Belgrade, Serbia, was successfully monitored during the 6-month follow-up period. Cardiac parameters, measured by conventional 2D echocardiography, were evaluated at baseline (3–5 days after MI) and 6 months after MI.

The control subjects were individuals who underwent a regular annual medical check-up at the Occupational Medical Centre, “Vinca” Institute of Nuclear Sciences, Belgrade, Serbia, during the period from February 2011 through December 2016. All of them underwent clinical, ultrasound, and electrocardiogram examination, and those with no evidence of cerebrovascular or cardiovascular diseases, chronic inflammatory diseases, T2DM, or renal failure were included in the study.

The study was approved by the ethics committees of the participating medical centers. All patients gave their written informed consent to participate in the study (approval code: NoIRB00003818, approved by Federal-Wide Assurance–FWA00006109; date of approval: 29 June 2011).

### 2.2. Doppler Echocardiography

The ultrasound procedure has been previously described [17]. Briefly, all Doppler echocardiographic studies were obtained using the commercially available, second harmonic imaging system Toshiba XG/Artida (Toshiba Medical Systems Corporation, Otawara City, Tochigi Prefecture, Japan), according to the American Society of Echocardiography and the European Association of Cardiovascular Imaging [20,21]. The examinations were performed at 2 check-points: within 3–5 days after admission and 6 months after MI.

The following parameters were measured or calculated: LV end-diastolic volume (LVEDV), LV end-systolic volume (LVESV), left atrial dimension (LAD), global radial strain (GRS), stroke volume (SV), and LV ejection fraction (LVEF), as previously described [17]. The change (Δ) in cardiac parameters during the 6 months was calculated as the difference between the value at the 6-month follow-up points and the baseline points.

The development of HF in patients 6 months after MI was diagnosed according to the 2012 European Society of Cardiology Guidelines for the diagnosis and treatment of HF, developed by a working group of the HF Association [22]. According to the severity of the patient’s symptoms and physical activity, HF was classified into functional categories: NYHA I-IV [23]. Patients in NYHA class I had no symptoms attributable to heart disease, while patients in NYHA classes II, III, or IV had mild, moderate, or severe symptoms, respectively. Advanced HF was defined as NYHA class > II [24].

### 2.3. Selection of Tag Variants

To maximize the coverage of genetic information, the Tagger server was used to select tag SNPs for genotyping, as described previously [18]. The selection criteria were: (i) SNPs located within the haplotype block containing the *LGALS-3* gene with a minor allele frequency (MAF) greater than 0.05; (ii) r^2^ > 0.8; (iii) select only the three best tags.

### 2.4. Genetic Analysis

Whole peripheral blood samples for DNA extraction were collected with EDTA within 3–5 days after MI. Genomic DNA was extracted by the phenol/chloroform extraction method [25]. Genotyping of rs4040064, rs11628437, and rs7159490 was done using TaqMan^®^ technology for allele discrimination, as previously described [18]. Approximately 10% of the samples were randomly selected and genotyped a second time by a different investigator. The results of the repeated genotyping were 100% consistent with the results of the original genotyping.

### 2.5. Reverse Transcription Quantitative Real-Time PCR (RT-qPCR)

Whole peripheral blood samples for total RNA extraction were collected from 100 patients in the 6-month follow-up points. Total RNA extraction from the PBMCs was performed within 30 min of collection using TRIzol^®^ Reagent (Invitrogen, Thermo Fisher Scientific, Waltham, MA, USA) according to the manufacturer’s instructions. The RNA samples were stored at −80 °C prior to use. Quantification of RNA was assessed using the BioSpec-nano spectrophotometer (Shimadzu Corporation, Kyoto, Japan). RNA integrity was evaluated by chip electrophoresis using the RNA 6000 Nano Kit (Agilent Technologies, Inc. Headquarters, Santa Clara, CA, USA) on the 2100 Bioanalyzer system (Agilent Technologies, Inc. Headquarters, Santa Clara, CA, USA). Of 100 samples collected, 92 yielded a total RNA of satisfactory quality with RNA Integrity Number (RIN) of 8–9 (range of 10 to 1) and were converted to cDNA.

Real-time PCR was performed in duplicate on an ABI Real-time 7500 system (Applied Biosystems, Foster City, CA, USA). Detection of *LGALS-3* gene expression was performed using the pre-developed TaqMan^®^ gene expression assay Hs00173587_m1 (Applied Biosystems, Foster City, CA, USA). Detection of the reference gene, *Peptidylprolyl isomerase A* (*Cyclophilin A*), was performed using the predeveloped TaqMan^®^ gene expression assay Hs99999904_m1 (Applied Biosystems, Foster City, CA, USA), as previously described [26].

### 2.6. Quantification of pGal-3 Levels

Whole peripheral blood samples, collected from 189 patients in the 6-month follow-up points, were centrifuged within 30 min of collection, and the resulting plasma samples were stored at −80 °C prior to use. The Elabscience^®^ Human GAL3 (galectin-3) ELISA Kit (Beijing, China) was used to determine Gal-3 concentrations according to the manufacturer’s instructions. Optical density was measured using the Perkin Elmer Wallac 1420 Victor2 Microplate Reader (PerkinElmer, Inc., Waltham, MA, USA) at 450 nm. A value representing the averaged duplicate readings of optical density measured for each sample was used to determine Gal-3 (ng/mL) using a four-parameter logistic (4 PL) curve-fit standard curve (My Assays: Online data analysis. Available at: https://www.myassays.com/four-parameter-logistic-curve.assay, accessed on 2 July 2021).

### 2.7. Statistical Methods

Allele frequencies and genotype distributions were estimated using the gene counting method. Differences in allele frequencies and genotype distributions between controls and patients, deviations from Hardy–Weinberg equilibrium (HWE), and comparisons of categorical variables were estimated using Pearson’s Chi-square (χ^2^) test. Means of normally distributed continuous variables were compared using the unpaired *t*-test, while medians of skewed continuous variables were compared using the non-parametric Mann–Whitney *U* test. Values of continuous variables were expressed as mean ± standard deviation (SD), and *p* values < 0.05 were considered statistically significant for the main demographic and biochemical parameters. Statistical analysis was performed using the software package Statistica Version 8 (StatSoft Inc, Tulsa, OK, USA) and IBM SPSS Statistics, Version 20.0 (IBM Corp., Armonk, New York, NY, USA).

Haplotype frequencies and their association with MI, changes in echocardiographic parameters within 6 months post-MI, plasma Gal-3 or *LGALS-3* mRNA levels, both 6 months post-MI, were performed using THESIAS 3.1 software [27,28]. The odds ratio (OR), with its 95% confidence interval (CI), was used as a measure of the strength of association between the haplotypes investigated and MI. The relative *LGALS-3* mRNA levels were standardized against the reference gene (*Cyclophilin A*) and presented as mean 2^−∆Ct^ values with their 95% CI, where ΔCt represents the difference between the Ct value for *LGALS-3* and Ct for *Cyclophilin A*, for each sample. According to the THESIAS software, the effects of haplotypes on *LGALS-3* mRNA expression, plasma Gal-3 levels and changes in echocardiographic parameters within 6 months post-MI are presented as expected phenotypic mean values for 1 dose of each haplotype with the corresponding 95% CI compared to the referent haplotype. The referent haplotype is set by the THESIAS software and represents the most frequent haplotype in the groups studied. The power of the study for the observed association of haplotypes with MI was calculated using the Power for Genetic Association Analyses (PGA) tool [29].

*p* values < 0.05 were considered statistically significant, except for the analysis of: (1) association of haplotypes with 6 different echocardiographic parameters (overall and according to gender); and (2) effect of haplotypes within the same study group (overall and according to gender). For these analyses, Bonferroni correction for multiple testing was performed, and *p* ≤ 0.005 and *p* < 0.017, respectively, were considered statistically significant.

## 3. Results

### 3.1. Association of rs4040064, rs11628437, and rs7159490 Haplotypes with MI

The general characteristics of the controls and the patients with the first MI are shown in Table 1. The patients had significantly higher BMI, triglycerides, and LDL cholesterol levels and lower HDL cholesterol levels than the control group. In addition, the patients with MI were older than the control subjects and had a higher proportion of males, hypertensives, and smokers.

The genotype and allele frequency distributions of rs4040064 G/T, rs11628437 G/A, and rs7159490 C/T in controls and patients are shown in Appendix A. The genotype and allele frequency distributions of the genetic variants investigated were in HWE in controls and patients (*p* > 0.05). The frequencies of the haplotypes inferred from rs4040064, rs11628437, and rs7159490 in the control group and the MI group, overall and subdivided by gender, are presented in Table 2. The haplotypes GGT, TAT, and TGT, with a frequency estimate <0.01 in at least one of the groups studied, were not included in the statistical analyses. In the haplotype analysis of variants rs4040064, rs11628437, and rs7159490, the GGC haplotype was set as a referent by the THESIAS software. Compared to it, the TGC haplotype was associated with a protective effect against MI in the group of males (OR [95% CI] = 0.19 [0.05–0.72], *p* = 0.015). The OR was adjusted for age, smoking, total cholesterol, triglycerides, and body mass index. We had the study power of 65% for the observed association at a significance level of *p* < 0.017. The same haplotype, TGC, had a frequency less than 0.01 in the group of female patients and in the patient group overall, so it was not included in further analyses. There was no significant association between the investigated haplotypes and MI in the group of females or in the study group overall. In addition, we analyzed the possible association of the haplotypes with BMI, when modelled as a continuous variable as well as a categorical variable (according to the presence of obesity: BMI < 30 kg/m^2^ obesity absent, BMI ≥ 30 kg/m^2^ obesity present) and found no significant association.

### 3.2. Relative LGALS-3 mRNA Expression in PBMCs Six Months Post-MI, in Regard to the Haplotypes Inferred from rs4040064, rs11628437, and rs7159490

Analysis of relative *LGALS-3* mRNA expression was performed on 92 patient samples, whose PBMCs were collected 6 months after the first MI. The general characteristics of this subgroup are shown in Appendix A. There was no significant association between the haplotypes investigated and relative *LGALS-3* mRNA expression in the MI patients overall (Figure 1A). In a group of males (N = 75), the GAT haplotype had a higher *LGALS-3* expression, compared to the referent haplotype (mean 2-∆Ct = 0.148 [0.091–0.205] vs. 0.088 [0.049–0.126], respectively, *p* = 0.05) (Revised Figure 1B). Values of *LGALS-3* expression in males did not differ statistically between patients according to diabetic status; therefore, we did not adjust the results for diabetic status. However, after Bonferroni correction for multiple testing, this association lost its significance. The haplotypes GGT, TAT, and TGT were excluded from the analysis due to the frequency estimate <0.01. In a group of females (N = 17), no significant association was found (Appendix A).

Quantitative real-time PCR was performed on 92 cDNA of human PBMCs to quantify the relative gene expression of *LGALS-3* and *Cyclophilin A*. Relative *LGALS-3* mRNA expression in regard to rs4040064, rs11628437, and rs7159490 variant haplotypes was assessed using THESIAS v3.1 software. Results are reported as the mean 2^−ΔCt^ value with its 95% confidence interval (CI) for each haplotype, where the ΔCt value is the difference between the Ct value of *LGALS-3* and the Ct value of *Cyclophilin A* for each sample. No significant association was found between the haplotypes inferred from rs4040064, rs11628437, and rs7159490 and relative *LGALS-3* mRNA expression in MI patients 6 months post-MI. The haplotypes GGT, TAT, and TGT with an estimate frequency <0.01 were not included in the statistical analyses. *^a^*—referent haplotype set by the Thesias software. *p* values were corrected for multiple testing, and values <0.017 were considered statistically significant.

### 3.3. Plasma Gal-3 in Patients Six Months Post-MI, in Regard to the Haplotypes Inferred from rs4040064, rs11628437, and rs7159490 and in Association with Echocardiographic Parameters Serving for Assessment of LV Function and Structure

Plasma Gal-3 concentrations were determined in 189 patients, whose PBMCs were collected 6 months post-MI. The general characteristics of this subgroup are shown in Appendix A. The median (25th, 75th percentile) Gal-3 plasma concentration was 15.73 ng/mL (7.84, 32.04 ng/mL). Based on the US Food and Drug Administration-cleared cut-off point for HF risk stratification (≤17.8 ng/mL low risk, 17.9–25.9 ng/mL intermediate risk, >25.9 ng/mL high risk), a threshold for plasma Gal-3 of 17.8 ng/mL was applied. The general characteristics of patients stratified into groups with plasma Gal-3 concentrations above and below 17.8 ng/mL, overall and divided by gender, are shown in Appendix A. In the present study, 82 patients (43.4%) had plasma Gal-3 concentrations above the upper limit of this threshold. Patients with higher plasma Gal-3 (>17.8 ng/mL) were more likely to be female, had higher BMI, and more severe HF (NYHA class III and IV) 6 months post-MI (Appendix A). Patients with NYHA class III and IV had significantly higher plasma Gal-3 levels, when modelled as a continuous variable, compared with patients with NYHA class I and II (50.8 ± 27.0 ng/mL vs. 21.8 ± 19.6 ng/mL, respectively, *p* = 0.001, Mann–Whitney *U* test) (Figure 2A). Patients with a higher plasma Gal-3 level (>17.8 ng/mL) also had a significantly higher increase in LAD within 6 months post-MI compared to those with a plasma Gal-3 level below this value (Appendix A). Changes in LVEDV and LVESV within 6 months post-MI were not significantly different between the 2 patient groups (Appendix A). Patients with systolic dysfunction 6 months post-MI, defined as LVEF < 40%, had significantly higher Gal-3 plasma levels, when modelled as a continuous variable, compared with those with LVEF ≥ 40% (30.2 ± 25.5 ng/mL vs. 22.6 ± 21.2 ng/mL, respectively, *p* = 0.03, Mann–Whitney *U* test) (Figure 2B).

Analyzing the association between the haplotypes investigated and plasma Gal-3 concentrations, we found that, in males (N = 142), carriers of the GAC haplotype had significantly higher plasma Gal-3 concentrations than carriers of the referent GGC haplotype, 6 months post-MI and adjusted for diabetic status (48.3 [37.3–59.4] ng/mL vs. 18.9 [14.5–23.4] ng/mL, respectively, *p* < 0.0001) (Table 3). We found no significant association between investigated haplotypes with plasma Gal-3 6 months post-MI overall (Table 3) or in a group of females (Appendix A).

Further, we evaluated the correlation between *LGALS-3* mRNA expression and plasma Gal-3 abundance in the PBMCs of a subset of 92 patients in whom both levels were measured, 6 months post-MI, independent of haplotypes. We found no correlation between *LGALS-3* mRNA and Gal-3 plasma levels overall (Spearman’s rank correlation coefficient 0.14; *p* = 0.20) nor in male patients only (Spearman’s rank correlation coefficient 0.11; *p* = 0.38).

### 3.4. Association of rs4040064, rs11628437, and rs7159490 Haplotypes with a Change in Echocardiographic Parameters Serving for Assessment of LV Function and Structure within Six Months Post-MI

We analyzed the possible association of rs4040064, rs11628437, and rs7159490 haplotypes with a change in echocardiographic parameters within 6 months (Δ values) in a subgroup of patients followed up for 6 months post-MI. Compared to the referent haplotype GGC, the haplotype GAT was associated with a change in LVESV (increase) and LVEF (decrease). After Bonferroni correction for multiple testing, only LVEF retained significance (Table 4). The haplotype GAT also showed a decrease in GRS and SV 6 months post-MI; however, these associations did not reach statistical significance (Table 4). The association of the haplotypes investigated with a change in echocardiographic parameters in males over time is presented in Appendix A. We found that the haplotype GAT was significantly associated with a decrease in LVEF in males, compared to the referent haplotype. These analyses for the female group were not presented due to the low number of subjects. However, there were no significant associations in this group.

## 4. Discussion

In the present study, we analyzed tag variants covering more than 80% of the variability of the haplotype block containing *LGALS-3*, in association with the first acute myocardial infarction, *LGALS-3* expression, and plasma Gal-3 concentrations 6 months post-MI. Three variants, namely, rs4040064, rs11628437, and rs7159490, were selected by the Tagger server [18] and previously analyzed in association with advanced carotid atherosclerosis [18]. The frequencies of the investigated alleles in the Serbian control group are consistent with those reported for the European, non-Finnish, population (subgroup: other non-Finnish European), according to the Genome Aggregation Database (gnomAD) [30]. So far, the genetic background of Gal-3 has been analyzed only sporadically [10,11,12]. One of the most investigated *LGALS-3* variants, the non-synonymous rs4652, was previously associated with Gal-3 levels [10] and is in an almost absolute LD with the variant investigated in the present study, rs11628437 (D′ = 1, r^2^ = 0.98). The variant rs4652 has previously been associated with CAD in diabetics [31] and systemic sclerosis, a disease characterized by chronic inflammation, like atherosclerosis [32]. To date, variants rs4040064 and rs11628437 have not been analyzed in association with any cardiovascular phenotype and are not present on microarray chips according to the Ensembl database [33]. The third variant investigated, rs7159490, has only been analyzed in GWASs, in association with ischemic stroke and stroke subtypes, drug-induced ventricular tachycardia Torsades de Pointes, and insulin resistance in childhood obesity [34,35,36,37]; however, no significant associations were observed with the phenotypes investigated. This makes our results valuable in determining the direction of future research.

In the present study, the mean age of the patients overall was 58.4 ± 11.4 years, and more than 70% of them were male. Since the statistics for heart disease based on the National Health and Nutrition Examination Survey (NHANES) 2011 to 2014 data show a higher incidence of myocardial infarction in males than females in this age group [38], it is clear why we had more males in our study group. Being aware of this and the higher Gal-3 concentrations in women compared to men, even in the healthy state [39,40], we performed all analyses in the study group overall and with regard to gender.

Analyzing the association of the haplotypes inferred from the variants rs4040064, rs11628437, and rs7159490 with the occurrence of the first acute MI, we found that the TGC haplotype was associated with a protective effect against MI in Serbian male patients, compared to the referent GGC haplotype, independent of confounding factors. In the MI group of females and overall, the TGC haplotype was not detected in the frequency required for this analysis and therefore the analysis could not be performed. This is the first association of this haplotype with the occurrence of the first MI and as such needs to be replicated and validated in a larger sample in order to accurately estimate the power of the association, particularly due to the lower frequencies in the Serbian population, which is reflected in the power of the study for this association of 65%. Since a significant increase in BMI with higher plasma levels of Gal-3 has been previously reported [41] and since we observed that patients with pGal-3 ≥ 17.8 ng/mL had higher BMI, we analyzed the possible association of haplotypes with BMI and found no significant association.

Further, in the study, we explored the haplotype effect of these variants on relative *LGALS-3* mRNA expression in PBMCs from 92 patients, 6 months after the first MI overall and divided by gender. We found that in the group of males, the GAT haplotype had higher *LGALS-3* mRNA expression, compared to the referent GGC haplotype. However, after Bonferroni correction for multiple testing, this association lost its significance. In carriers of the same GAT haplotype, there was a significant decrease in LVEF within 6 months post-MI, compared to the referent haplotype, in the group of males and overall. Data from the FIVEx eQTL catalogue show a significant increase in *LGALS-3* expression in peripheral blood, aorta, and coronary artery by the rs4040064 G allele, in peripheral blood and tibial artery by the rs11628437 A allele, and in tibial and coronary arteries by the rs7159490 T allele [15]. Previously, we had also found no significant association between the haplotypes investigated and *LGALS-3* expression in patients with carotid atherosclerotic plaque, but the analysis was conducted on only 39 tissue specimens [18].

We quantified plasma Gal-3 concentrations in 189 patients whose PBMCs were collected 6 months after the first MI and demonstrated a strong association of haplotype GAC with plasma Gal-3 levels in males, compared to the referent GGC haplotype, independent of diabetic status. Very recently, a genome-wide meta-analysis of 90 cardiovascular-related proteins (including Gal-3) in 30,931 individuals reported an association of 5 protein quantitative trait loci (pQTLs) that affect circulating Gal-3 and were presented as cis-regulators, independently of MI. Wild type alleles of two of them, rs76480089 A/G and the low-frequency (in the European, non-Finnish, population) variant rs6650508 A/G, are in an almost absolute (D′ = 1, r^2^ = 0.90) and absolute LD (D′ = 1, r^2^ = 1) with the rs7159490 C allele, respectively, and are associated with higher expression of Gal-3 in peripheral blood [12]. Another GWAS of 83 plasma protein biomarkers in cardiovascular disease (including Gal-3) in 3711 individuals with at least 3 established cardiovascular disease risk factors, but without history or symptoms of prevalent disease at baseline, reported an association of 4 pQTLs that affect plasma Gal-3 levels [11]. Three SNPs (namely, rs33988101, rs7928577, and rs1169306) are all acting over distances greater than 100 MB or at different chromosomes from *LGALS-3*, so they were presented as trans-regulators. The wild type allele of a fourth one, rs9323280 A/C, is in an almost absolute LD with the rs7159490 C allele (D′ = 1, r^2^ = 0.95). Using the data from the public GWA studies, the authors further evaluated the abovementioned SNPs for their causal effect on CAD and found that all four were associated with CAD risk. These data are consistent with our findings that the GAC haplotype inferred from the variants tagging *LGALS-3*-containing haplotype block is significantly associated with elevated plasma Gal-3 levels in males.

We also wanted to evaluate whether there was a correlation between plasma Gal-3 and *LGALS-3* mRNA levels in a subset of patients in whom both expressions were measured, overall (N = 92) and in male patients only (N = 73). The correlation did not reach statistical significance. Apart from the low number of patients for this analysis, it is known that correlations between mRNA and protein abundance are poor and rarely reflect each other ex vivo in human tissue. This discordance is attributed to multiple biological mechanisms that contribute to establishing the expression level of a protein beyond mRNA concentration. These include: (1) translation rates, which can be modulated through the binding of regulatory proteins/non-coding RNAs or through the ribosome occupancy/availability, (2) modulation of a protein’s half-life, (3) delay of protein synthesis, and (4) protein transport [42,43].

In the early period following acute MI, Gal-3 is an essential factor in the onset and development of the protective, healing, fibrogenic process of the damaged MI, in order to maintain the geometry and function of the heart [44,45]. However, continuous and prolonged inflammation and collagen production, along with high systemic and cardiac Gal-3 levels, lead to tissue fibrosis and accelerate unfavorable cardiac remodeling and further development of HF [46]. Therefore, measuring Gal-3 levels in the acute MI phase during the early stages may not be a reliable marker in terms of the further remodeling process. 

In the present study, we measured plasma Gal-3 6 months after MI with the idea to overcome the acute-phase response that could blunt the analysis of a possible association between tag variants and Gal-3 expression. Beyond that, we wanted to analyze it in association with the change in echocardiographic parameters of LV function and structure within 6 months post-MI and CAD severity in Serbian patients. Regarding the long-term outcomes following MI, elevated Gal-3 (3, 6, or 12 months post-MI) was associated with re-hospitalization and mortality, compared with patients with Gal-3 levels that decreased or remained stable over time [47,48]. In our study, the median plasma Gal-3 was 15.73 ng/mL (7.84, 32.04 ng/mL), and nearly 45% of patients had plasma Gal-3 above the upper limit cut-off value for HF risk (17.8 ng/mL). Drug usage did not affect *LGALS-3* mRNA expression or plasma levels of galectin-3 in the patient group. We found that higher plasma Gal-3 levels (>17.8 ng/mL) 6 months post-MI were significantly associated with a higher NYHA functional class, reflecting the severity of HF in patients overall. Patients with higher plasma Gal-3 (>17.8 ng/mL) 6 months post-MI also had a greater increase in left atrial dimension within 6 months post-MI in patients overall; however, the change in LVEDV and LVESV within 6 months did not correlate with plasma Gal-3. Pecherina et al. found an increase in serum Gal-3 at both 6 and 12 months after MI, but no significant correlations were demonstrated with echocardiographic parameters serving for assessment of LV function and structure [49]. In our study, plasma Gal-3 levels were significantly higher in patients overall with systolic dysfunction 6 months post-MI, defined as LVEF < 40%, than in patients with LVEF ≥ 40%. Similar to our result, Mayr et al. showed that patients with plasma Gal-3 concentrations above the median value of 10.86 ng/mL 4 months after the MI presented with significantly impaired LVEF [50].

This study has some limitations that need to be addressed and considered for future research. The number of females in the patient group was rather small to conduct an accurate and valid analysis. Since scientific data published in the meantime has identified some other SNPs as eQTL or pQTL that are not located in this haplotype block, these should be included in future analysis along with a valid LD matrix analysis. The haplotype-based approach, especially of tag variants, has a larger effect size and provides a more accurate estimation of the possible associations of genetic variants with the phenotype of interest than the single-SNP approach. However, in order to more accurately estimate the strength of the association of the haplotypes investigated with the incidence of myocardial infarction, *LGALS-3* mRNA levels and protein expression, replication, and validation in a larger sample size is needed.

In conclusion, we have shown that the GAC haplotype inferred from the 3 variants tagging the *LGALS-3*-containing haplotype block bears a strong association with plasma Gal-3 levels 6 months post-MI in Serbian male patients and suggest that the TGC haplotype may have a protective effect against the occurrence of MI in the same group. Investigated variants have not shown significant associations with the change in LV echocardiographic parameters within 6 months. Therefore, analyzed variants could be promising haplotype-tagging variant markers for plasma Gal-3 levels in CAD patients post-MI.

## Figures and Tables

**Figure 1 genes-14-00109-f001:**
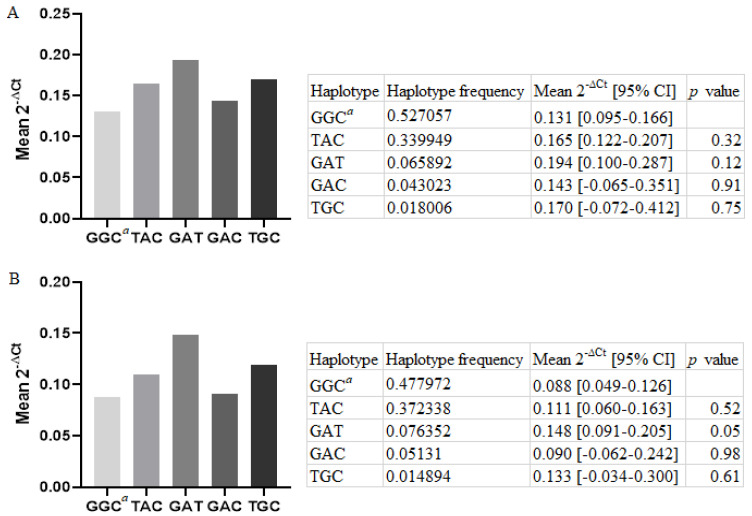
Relative *LGALS-3* mRNA expression in PBMCs from patients 6 months post-MI in regard to the haplotypes of the variants rs4040064, rs11628437, and rs7159490, (**A**) overall and (**B**) in males.

**Figure 2 genes-14-00109-f002:**
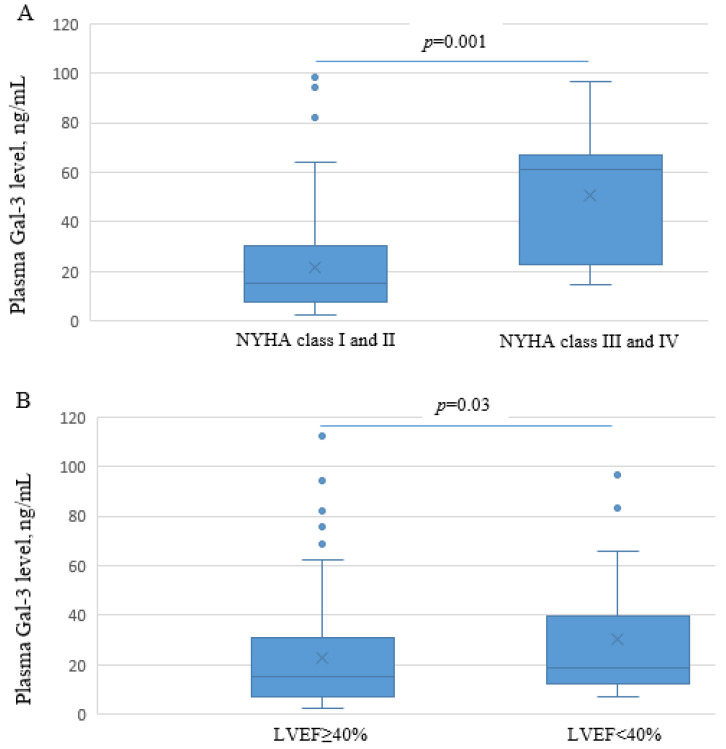
Plasma Gal-3 levels in patients 6 months post-MI according to the: (**A**) NYHA class 6 months after MI and (**B**) systolic dysfunction 6 months post-MI, defined as LVEF < 40%. (**A**) Patients with NYHA class III and IV had significantly higher plasma Gal-3 levels than patients with NYHA class I and II (50.8 ± 27.0 ng/mL vs. 21.8 ± 19.6 ng/mL, respectively, *p* = 0.001, Mann–Whitney *U* test). (**B**) Patients with systolic dysfunction 6 months post-MI, defined as LVEF < 40%, had significantly higher Gal-3 plasma levels compared with patients with LVEF ≥ 40% (30.2 ± 25.5 ng/mL vs. 22.6 ± 21.2 ng/mL, respectively, *p* = 0.03, Mann–Whitney *U* test).

**Table 1 genes-14-00109-t001:** General characteristics of controls and patients with first acute myocardial infarction.

Variable	Control Group, N = 323	MI Group, N = 546	*p* Value
Age, years	54.1 ± 14.2	58.4 ± 11.4	<0.01 ^§^
Gender, f/m, %	44.58/55.42	28.39/71.61	<0.01 ^§^
BMI, kg/m^2^	25.09 ± 3.68	27.26 ± 4.01	<0.01 ^§^
TC, mmol/L	5.61 ± 1.30	5.61 ± 1.51	ns
HDLC, mmol/L	1.48 ± 0.86	1.12 ± 0.34	<0.01 ^§^
LDLC, mmol/L	3.31 ± 1.23	3.67 ± 1.04	<0.01 ^§^
TG, mmol/L	1.58 ± 1.09	1.86 ± 1.27	<0.01 ^§^
T2DM, %	0.00	17.10	N/A
Hypertension, %	27.43	65.76	<0.01
Current smokers, %	55.04	64.09	0.06

Values are mean ± SD for: body mass index (BMI), age, total cholesterol (TC), triglycerides (TG), high-density lipoprotein cholesterol (HDLC), and low-density lipoprotein cholesterol (LDLC). ^§^ Mann–Whitney *U* test was used to compare the values of continuous variables with a skewed distribution between controls and MI patients. Pearson’s Chi-square (χ^2^) test was used for comparison of the categorical variables. *p* values < 0.05 were considered statistically significant. T2DM: type 2 diabetes mellitus; ns: not significant; N/A: not applicable.

**Table 2 genes-14-00109-t002:** Haplotype effects of the variants rs4040064, rs11628437, and rs7159490 on risk of myocardial infarction, overall and divided by gender.

Haplotype ^§^	Haplotype Frequency		Haplotype Effect on Risk of MI	
	Control Group	MI Group	OR [95% CI] ^#^	*p* Value
**Overall**	N = 323	N = 546		
GGC	0.61625	0.6113	referent haplotype	
TGC ^¥^	0.01667	0.00437	N/A	N/A
TAC	0.25505	0.25879	1.06 [0.83–1.37]	0.62
GAT	0.07233	0.0886	1.30 [0.85–1.98]	0.22
GAC	0.01799	0.02834	1.69 [0.77–3.71]	0.19
TGT ^¥^	0.00953	0.00146	N/A	N/A
TAT ^¥^	0.0074	0.00223	N/A	N/A
GGT ^¥^	0.00478	0.00492	N/A	N/A
**Males**	N = 179	N = 391		
GGC	0.59517	0.58863	referent haplotype	
TGC	0.03082	0.00543	0.19 [0.05–0.72]	0.015
TAC	0.2674	0.27426	1.12 [0.80–1.57]	0.49
GAT	0.06479	0.09247	1.54 [0.87–2.72]	0.13
GAC	0.01739	0.02933	1.79 [0.64–4.50]	0.27
TGT ^¥^	0.01083	0.00193	N/A	N/A
TAT ^¥^	0.00772	0.00278	N/A	N/A
GGT	0.00587	0.00517	0.99 [0.10–9.64]	0.99
**Females**	N = 144	N = 155		
GGC	0.63392	0.67037	referent haplotype	
TGC ^¥^	0.00507	0.00022	N/A	N/A
TAC	0.2443	0.22015	0.88 [0.58–1.35]	0.56
GAT	0.0856	0.07816	0.90 [0.46–1.76]	0.75
GAC	0.01915	0.02566	1.25 [0.33–4.66]	0.74
TGT ^¥^	0.00448	0.0013	N/A	N/A
TAT ^ǂ^	0.00	0.00	N/A	N/A
GGT ^¥^	0.00749	0.00414	N/A	N/A

^§^ The alleles in haplotypes are in the following order: rs4040064 G/T, rs11628437 G/A, and rs7159490 C/T. ^¥^ Haplotypes with a frequency estimate <0.01 in both groups, controls and MI patients, were not included in statistical analyses. Haplotypes with a frequency estimate >0.005 were considered relevant for the analysis. ^ǂ^ According to the THESIAS software, the TAT haplotype has not been detected in an analyzed group of controls and patients or in the group of females. ^#^ The OR was adjusted for age, smoking, total cholesterol, triglycerides, and body mass index. *p* values were corrected for multiple testing, and values < 0.017 were considered statistically significant. OR: odds ratio; CI: confidence interval; N/A: not applicable.

**Table 3 genes-14-00109-t003:** Plasma Gal-3 levels in patients 6 months post-MI according to the haplotypes inferred from the variants rs4040064, rs11628437, and rs7159490, overall and in males, adjusted to diabetic status.

Haplotype ^§^	Haplotype Frequency	Mean pGal-3 [95% CI], ng/mL	*p* Value
Overall, N = 189			
GGC	0.594489	41.2 [36.3–46.2]	referent haplotype
TAC	0.285885	43.3 [35.1–51.5]	0.68
GAT	0.06574	45.7 [31.4–59.9]	0.59
GAC	0.03387	55.2 [34.5–75.9]	0.19
Males, N = 142			
GGC	0.529661	18.9 [14.5–23.4]	referent haplotype
TAC	0.327339	17.2 [10.7–23.7]	0.69
GAT	0.074224	19.3 [3.8–34.8]	0.96
GAC	0.041146	48.3 [37.3–59.4]	<0.0001

^§^ The alleles in the haplotypes are in the following order: rs4040064 G/T, rs11628437 G/A, and rs7159490 C/T. *p* values were corrected for multiple testing, and values <0.017 were considered statistically significant.

**Table 4 genes-14-00109-t004:** Association of rs4040064, rs11628437, and rs7159490 haplotypes with a change in echocardiographic parameters within 6 months post-MI.

Cardiac Parameter	Haplotype Frequency	Means [95% CI]	*p* Value
Δ LV End diastolic volume (mL)			
GGC	0.579682	5.78 [0.01–11.55]	ref. haplotype
TAC	0.292608	−2.72 [−13.81–8.37]	0.23
GAT	0.070432	6.12 [−15.85–28.09]	0.98
GAC	0.035608	−11.09 [−42.46–20.28]	0.3
Δ LV End systolic volume (mL)			
GGC	0.579747	3.24 [−0.78–7.27]	ref. haplotype
TAC	0.292733	−2.98 [−11.12–5.16]	0.22
GAT	0.070111	17.10 [4.18–30.02]	0.04
GAC	0.035804	−11.49 [−43.27–20.29]	0.37
Δ LV Ejection fraction (%)			
GGC	0.582037	1.13 [−0.54–2.80]	ref. haplotype
TAC	0.28883	3.22 [0.07–6.36]	0.31
GAT	0.072803	−6.31 [−11.15–−1.47]	0.005
GAC	0.035033	8.83 [−1.99–19.65]	0.17
Δ Left atrial dimension (mm)			
GGC	0.582591	1.90 [0.97–2.83]	ref. haplotype
TAC	0.287416	0.32 [−1.73–2.37]	0.21
GAT	0.073188	−0.86 [−6.65–4.94]	0.36
GAC	0.035369	−1.01 [−16.96–14.93]	0.72
Δ Global radial strain (%)			
GGC	0.59579	4.12 [1.76–6.48]	ref. haplotype
TAC	0.281092	−1.01 [−5.60–3.59]	0.09
GAT	0.069458	−3.30 [−10.43–3.83]	0.06
GAC	0.040206	4.61 [−7.89–17.11]	0.94
Δ Stroke volume (mL)			
GGC	0.584989	8.03 [3.38–12.68]	ref. haplotype
TAC	0.290219	0.78 [−8.12–9.67]	0.2
GAT	0.067594	−13.50 [−35.46–8.46]	0.06
GAC	0.034273	−7.99 [−32.82–16.83]	0.21

Values are shown as expected phenotypic means with their 95% confidence interval (CI). *p* values were corrected for multiple testing, and values ≤0.005 were considered statistically significant. Δ: Change from 3–5 days to 6 months.

## Data Availability

The data presented in this study are contained within the article and Appendix A.

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
