# Peer review of "Variants Tagging LGALS-3 Haplotype Block in Association with First Myocardial Infarction and Plasma Galectin-3 Six Months after the Acute Event"

_genes, 2022, doi:10.3390/genes14010109_

Round 1

Reviewer 1 Report

The manuscript is interesting and has novelty. However, it needs some major revisions as follow:    1. It is best to avoid using abbreviations and acronyms in the title and abstract.  2. Structured abstract is needed. An abstract with distinct, labeled sections (e.g., Introduction, Methods, Results, Conclusion). 3. Grouping should express structured in a logical way to make them easier to follow. 4. It is not necessary to mention the details of the protocol in the methods. The details for RT-qPCR and Quantification of pGal-3 levels are not necessary in this manuscript. They are quite standardized and are not informative in terms of the objective in this study. 5. The authors should have considered the genetic ancestry as the potential confounders in the statistical analysis. 6. Authors should report their comment regarding the fact that ESC recommends to avoid genetic testinf in CVD apart from limiting cohorts of patients. 7. Cost of the implementation of the findings into practice. Please, give your opinion about it. 8. This is the hospital-based case-control study, which is prone to selection bias. I would recommend the authors’ thoughts  about the potential selection bias in this study in the Discussion section.  9. The entire manuscript would benefit significantly from a grammatical revision.

Reviewer 2 Report

The study "Variants tagging LGALS-3 haplotype block in association with first MI and plasma Galectin-3 six months post-MI" is very interesting and brings new information about the role of haplotypes of the LGALS-3 gene in plasma levels of Galectin-3 in patients six months after the first myocardial infarction. The manuscript is well written, however authors must take care at conclusions, since it seems premature to assume that the TGC haplotype "confers protection against myocardial infarction in men" based only on the frequency of the haplotype. In this way, I recommend the manuscript for publication after some justifications and the minor corrections described below:

1- Authors must correct the abstract according to the journal's instructions in a single paragraph, with a maximum of 200 words;

2- Include in the header of the tables accompanying figures 1A and 1B – Mean 2-ΔCt [95% CI] and include in the legend of figures 1A and 1B the meaning of the symbol in the GGC haplotype;

3- Include the sample number for each haplotype in the tables so that readers have an idea of the sample size analyzed per haplotype;

4- The authors performed a correlation analysis between LGALS-3 mRNA expression and plasma levels of Gal-3 in 92 patients and found no correlation. For this analysis, were the data stratified by haplotype and by gender? If so, the authors should comment on the results since this study shows a difference between the GAC haplotype and plasma levels of Gal-3 only in male patients;

5- It is premature to assume that the TGC haplotype conferred protection against MI in males. None of the other studied parameters showed the TGC haplotype. For example, the expression of the LGALS-3 gene in the TGC haplotype was not different from the reference haplotype or from the other haplotypes, even in men. In addition, plasma Gal-3 levels were not shown in individuals with the TGC haplotype, and this could have been included in Table 3. Do the authors have plasma Gal-3 data from the control group with the TGC haplotype? Please justify.

6- The authors need to justify how the GAC haplotype in men is associated with an increase in plasma Gal-3, while the GAT haplotype, in addition to presenting higher gene expression, was also associated with more unfavorable echocardiographic parameters such as an increase in LVSV and a reduction in LVEF (%)? Which of the two haplotypes would be more unfavorable for the patient with AMI and why? please justify

7- Since supplementary table 5 shows the significant increase in BMI due to higher plasma levels of Gal-3, and that some studies claim that there is an association between plasma levels of Gal-3 and BMI (for example: J Am Heart Assoc 2022 May 3;11(9):e023238.doi:10.1161/JAHA.121.023238), I would like to know if the authors evaluated a possible correlation between haplotypes and BMI in patients with IM, or if they commented on the discussion about the influence of BMI in the present study;

8- The authors do not mention a possible interference of the drugs used by the patients in the expression of the LGALS-3 gene and especially in the plasma levels of Gal-3. Has this possibility ever been evaluated? Please comment.
